# Emergency department visits due to hand trauma and subsequent emergency hand surgery in three Finnish hospitals during the first and second waves of COVID-19 pandemic

Ilari Kuitunen[1,2]*, Jarkko Jokihaara[3,4], Ville Ponkilainen[5], Aleksi Reito[5], Juha Paloneva[1,6], Ville M. Mattila[4,5], Antti P. Launonen[5]

1 School of Medicine, University of Eastern Finland, Kuopio, Finland, 2 Mikkeli Central Hospital, Mikkeli, Finland, 3 Department of Hand- and Microsurgery, Tampere University Hospital, Tampere, Finland, 4 Faculty of Medicine and Health Technologies, Tampere University, Tampere, Finland, 5 Department of Orthopaedics and Traumatology, Tampere University Hospital, Tampere, Finland, 6 Central Finland Hospital, Jyväskylä, Finland

* ilari.kuitunen@uef.fi

**Data Availability Statement:** Data cannot be shared publicly because of Finnish register

## Abstract

### Introductions

The rate of acute hand trauma visits to emergency departments (ED) and surgeries decreased during the COVID-19 lockdown. Our aim was to analyze the influence of national lockdown during the first wave and the regional restrictions during the second wave on the rate of visits to the ED and urgent hand surgeries in Finland.

### Methods

Material for this retrospective study was gathered from three Finnish hospitals All ED visits and urgent or emergency surgeries from January 2017 to December 2020 were included. Incidences per 100 000 persons with 95% confidence intervals (CI) were calculated and compared by incidence rate ratios (IRR).

### Results

The incidence of hand injury was lower after the beginning of the lockdown in March 2020 (IRR 0.70 CI 0.63–0.78). After lockdown ended in May, the monthly incidences of ED visits returned to the reference level. During the lockdown, the incidence of fractures and dislocations was 42% lower in March (IRR 0.58 CI 0.50–0.68) and 33% lower in April 2020 (IRR 0.67 CI 0.57–0.80). The incidence of fracture repair surgeries was 43% lower in March 2020 (IRR 0.57 CI 0.35–0.93) and 41% lower in July 2020 (IRR 0.59 CI 0.36–0.98). Incidence of replantation was 49% higher in March 2020 (IRR 1.49 CI 0.53–4.20) and 200% higher in July 2020 (IRR 3.00 CI 0.68–13.2) but these increases had high uncertainty.

legislation. Data are available from the participating hospitals Institutional Data Access Committee for researchers who meet the criteria for access to confidential data as this data contains sensitive information. Research permission requests to obtain the access to data can be submitted to satu. yla-mononen@pshp.fi in Tampere University Hospital, paivi.lampinen@ksshp.fi in Central Finland Hospital and pirkko.tikkanen@essote.fi in Mikkeli Central Hospital.

**Funding:** The authors received no specific funding for this work.

**Competing interests:** The authors have declared that no competing interests exist.

## Conclusions

The rate of ED visits due to hand injuries decreased while the rate of emergency hand operations remained unchanged during the national COVID-19 lockdown in spring. After the lockdown, the incidences returned to reference level and were unaffected by regional restrictions during the second wave of pandemic.

## Introduction

On March 12, 2020, the World Health Organization declared COVID-19 a global pandemic [1]. In Finland, the Government declared a state of emergency and nationwide lockdown and implemented several measures to enforce social distancing on March 16. These measures included a ban on social gatherings of more than 10 persons, the closure of external borders, and Finnish citizens returning from abroad were ordered to stay in quarantine at home for two weeks. In addition, public institutions, including primary schools, were closed and working remotely from home, where possible, was encouraged. These restrictions remained in force until the end of May 2020 [2]. As the second wave began in September 2020 in Finland, were regional stepwise restrictions used instead of nationwide lockdown.

Although surgical societies have provided guidelines on how to treat and operate surgical patients with COVID-19 [3,4], the effects of social restrictions on the demand for surgical care is not well understood. Following the start of the nationwide lockdown in the UK in March 2020, the rate of hand traumas referred to the Hand Trauma Clinic in London decreased by 75%, after which rates slowly returned to normal during April [5]. Another recent French study from Paris reported that although the overall rate of hand injuries decreased during the lockdown, domestic hand injuries increased when compared to the rate in 2019, whereas work-related injuries decreased. Moreover, a larger proportion of the patients admitted with hand injuries in Paris required operative treatment in 2020 (52%) compared to the corresponding dates in 2019 (37%) [6]. Trauma operations decreased by 30% in Finnish children during the first wave of the pandemic, but the decrease was mainly due to reduced rates of lower limb injuries, whereas the rate of upper limb operations remained nearly unchanged compared to pre-pandemic era [7]. The association between lockdown and change in the risk for hand injury is not straightforward because the majority of hand injuries occur in domestic or other everyday situations, and therefore the rate did not slow down during the lockdown [8].

The aim of our study was to describe the influence of the national lockdown, social distancing, regional restrictions and remote working on the number of visits to emergency departments and emergency or urgent hand surgeries performed during the first and second waves COVID-19 pandemic in Finland. The identification of changes in injury patterns and the subsequent demand for surgical care are important factors in the planning of the optimal use of resources during a national state of emergency.

## Materials and methods

The data for this multicenter, retrospective study were collected from three Finnish hospitals that provide primary, secondary and tertiary care. These hospitals cover a total catchment area of 900 000 residents, which accounts for 1/6th of the Finnish population [9]. Moreover, the catchment area for replantation and revascularization surgery at these hospitals comprises a

population of over three million residents (more than half of the population). Thus, the three hospitals are representative of the Finnish population as a whole. Data on all visits to the emergency department (ED) due to hand injuries and all emergency or urgent (operated within 7 days after referral) hand operations were collected from 1.1.2017 to 31.12.2020. We included patients 16 years of age and more at the time of the injury.

Hand injury was defined as an injury to the distal forearm, wrist, or hand. The Finnish version of the Nomesco surgical procedural codes 10 [10], ICD-10 diagnosis codes, and patient characteristics were collected from the electronic patient information systems of the participating hospitals using data management software. (Table 1) Hand injuries were further classified as minor injuries (skin wounds, sprains, bruises), fractures and dislocations, tendon injuries (flexor or extensor), or major injuries (injuries requiring emergency vascular repair or replantation). The healthcare system in Finland is publicly funded and accessible for all emergency patients. All emergency patients requiring hand surgery have access to operative treatment in public hospitals. We retrieved daily number of positive COVID-19 cases from the open access data of the Finnish Institute of Health and Welfare, available online from ww.thl.fi/en.

Clinical and demographic data have been presented as means and standard deviations (SD) or as counts and percentages. Monthly incidences with 95% confidence intervals (CI) per 100 000 persons were calculated by using the Poisson exact method. Data from the year 2020 were compared with reference years 2017–2019 by using incidence rate ratios (IRR). The analyses were performed using R version 4.0.4 (R Foundation for Statistical Computing, Vienna, Austria).

**Table 1. Included operations based on the Nomesco surgical procedure classification and included diagnoses based on the international classification of diseases-10.**

| Procedure code | Explanation |
| --- | --- |
| NDP10 | Replantation of hand |
| NDP12 | Replantation of a digit |
| NDP18 | Replantation of several digits |
| NDP30 | Repair of wrist or hand by transplant of tissue |
| NDP32 | Repair of finger or fingers by transplant of tissue |
| NDL30 | Suture or reinsertion of tendon of wrist or hand, flexor tendon |
| NDL32 | Suture or reinsertion of tendon of wrist or hand, extensor tendon |
| NDL34 | Suture or reinsertion of tendon of wrist or hand, other tendon |
| NDJ60 | Internal fixation of fracture of wrist or hand with screw |
| NDJ62 | Internal fixation of fracture of wrist or hand using plate and screws, scaphoid |
| NDJ64 | Internal fixation of fracture of wrist or hand wire, rod, cerclage or pin |
| Diagnose code | |
| S52.5 | Fracture of lower end of radius |
| S52.6 | Fracture of lower end of both ulna and radius |
| S60.0 –S60.9 | Superficial injury of wrist and hand |
| S61.0 –S61.8 | Open wound of wrist and hand |
| S62.0 –S62.8 | Fracture at wrist and hand level |
| S63.0 –S63.9 | Dislocation, sprain and strain of joints and ligaments at wrist and hand level |
| S64.0 –S64.9 | Injury of nerves at wrist and hand level |
| S65.0 –S65.9 | Injury of blood vessels at wrist and hand level |
| S66.0 –S66.9 | Injury of muscle and tendon at wrist and hand level |
| S67 | Crushing injury of wrist and hand |
| S68.0 –S68.9 | Traumatic amputation of wrist and hand |
| S69 | Other and unspecified injuries of wrist and hand |

## Ethics

According to the Finnish research legislation and The Finnish National Board on Research Integrity, appointed by the Ministry of Education and Culture: "The review of the ethics committee is not required for the research of public and published data, registry and documentary data and archive data.". https://tenk.fi/en/advice-and-materials/guidelines-ethical-review-human-sciences. The Ethics Committee of Tampere University Hospital has waived ethical evaluation of all register-based studies, in which the participants are not contacted. https://www.tays.fi/en-US/Research_and_development/Ethics_Committee. Institutional permissions were obtained from Chief doctors of each of the participating hospitals to access the hospital discharge register data. Informed consent from patients are not needed when retrospective register data is handled and the participants are not contacted.

## Results

During the four-year study period, a total of 32 506 hand injuries were treated in the ED of the participating hospitals, and 2 474 emergency or urgent hand operations were performed. Of these, 6 477 (19.9%) ED visits and 509 (20.5%) surgeries occurred during the pandemic period (from March 2020 to December 2020). The incidence of hand injury was lower before the start of the lockdown in February 2020 when compared with reference years (IRR 0.88 CI 0.78–0.98; Fig 1A), and after the beginning of the lockdown in March 2020 the incidence of ED visits decreased (IRR 0.70 CI 0.63–0.78; Fig 1A). Correspondingly, after the end of May and lockdown, the incidence increased and the monthly incidence peaked and the highest monthly incidence within the study period was reported in June 2020, 98 ED visits per 100 000 person-months (IRR 1.12 CI 1.02–1.24; Fig 1A). After June 2020, the monthly incidences of ED visits due to hand injuries remained at the reference level (Fig 1A). The age and gender distribution of ED patients remained unchanged in year 2020 (Table 1).

 The most common reasons for ED visit were minor hand injuries followed by fractures and dislocations of the hand or wrist. The incidences of minor injuries were 16% lower in March during the lockdown (IRR 0.84 CI 0.71–0.99; Fig 1B). The most prominent change during the lockdown was seen in the incidence of fractures and dislocations which was 42% lower in March (IRR 0.58 CI 0.50–0.68) and 33% lower in April 2020 (IRR 0.67 CI 0.57–0.80) than in the reference years (Fig 1B). After the lockdown both minor and major injuries shifted back to the level of reference years (Fig 1B).

 The overall incidence trend of emergent and urgent hand operations in 2020 was similar to reference years Fig 2A. Men were more likely to sustain hand injury requiring operative treatment (Table 2). The lockdown did not increase the proportion of patients waiting for operation over 48 hours after the referral (Table 2). During the lockdown the rate of replantation was 49% higher in March 2020 than in the reference years (IRR 1.49 CI 0.53–4.20; Fig 2B), and in July 2020, the rate of replantation was 200% higher (IRR 3.00 CI 0.68–13.2; Fig 2B), but these findings have high uncertainty. On the contrary, the incidence of fracture repair was 43% lower in March 2020 (IRR 0.57 CI 0.35–0.93; Fig 2B) and 41% lower in July 2020 (IRR 0.59 CI 0.36–0.98; Fig 2B). The incidences of tendon repairs in 2020 were similar to reference years (Fig 2B).

 The daily number of positive COVID-19 findings is presented in Fig 3. During the first wave the testing capacity was limited, but during the second way testing capacity was in full use.

## Discussion

The national COVID-19 lockdown seemed to have a clear decreasing impact on the rate of visits to the ED due to hand injuries during the first wave of COVID-19. The decrease was mainly

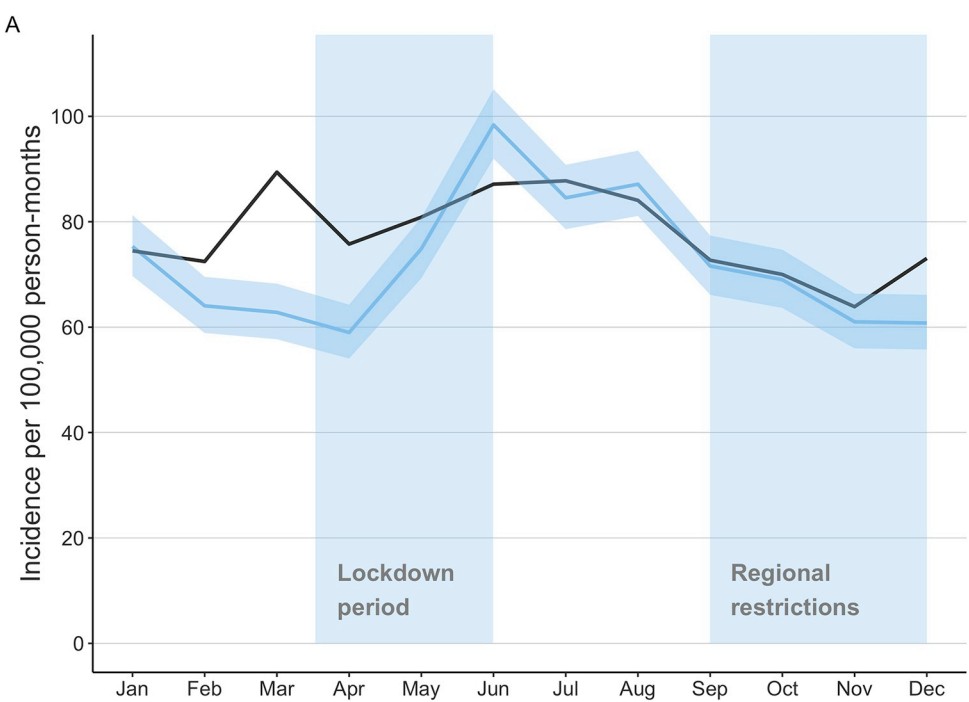

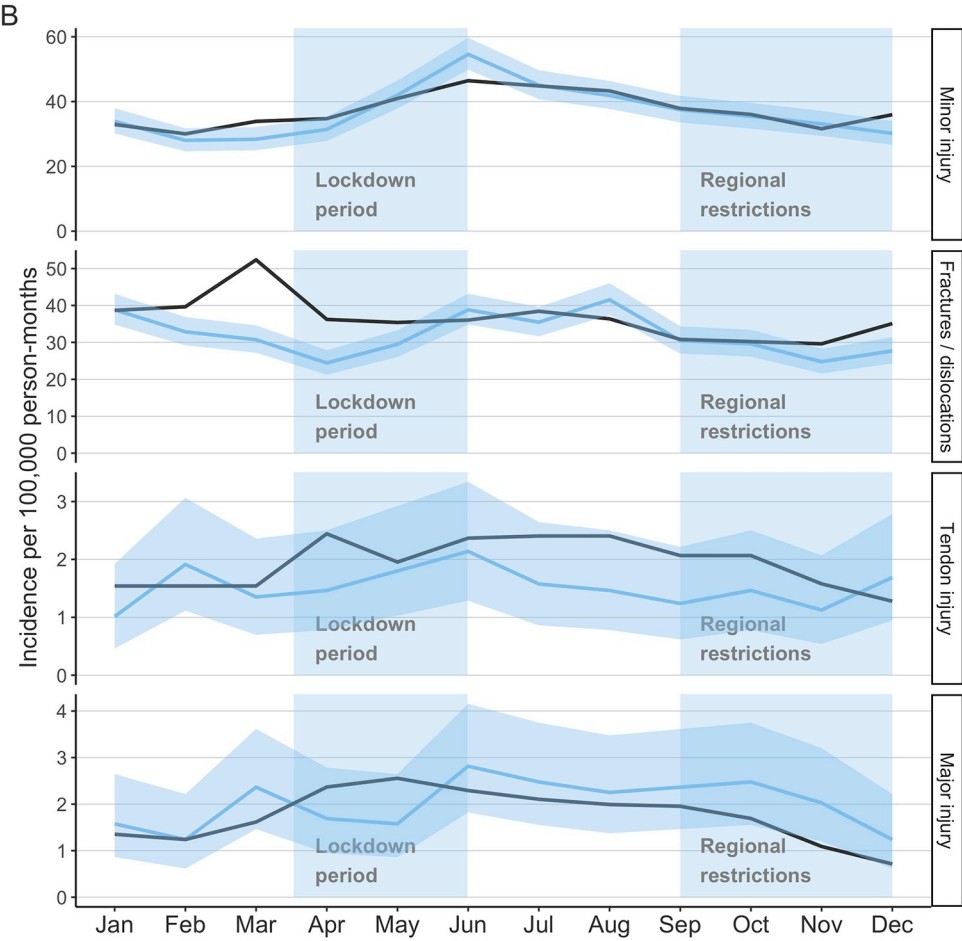

**Fig 1. A** Monthly incidences of emergency department visits due hand injuries. Blue line with 95% confidence intervals (light blue) presents year 2020 and black line presents the average incidence for reference years 2017–2019. **B** Monthly incidences of emergency department visits due hand injuries stratified by the diagnose of the visit. Blue line with 95% confidence intervals (light blue) present year 2020 and black line presents the average incidence for reference years 2017–2019.

due the reduced number of fractures and dislocations during the lockdown as the other injury types remained unchanged. The overall rate of emergency or urgent hand operations remained unchanged during the lockdown in Finland, despite the clear decrease in the rate of fracture repairs. A high, but uncertain, increase was seen in the number of replantation or revascularization operations after the start of the lockdown. However, this increase may be a non-specific change, and a similar temporal variation was also observed in the data from the reference years. The rate of revascularizations was again higher after the lockdown during the summer, when restrictions were lifted, but also this finding had high uncertainty. When compared to the major decreases seen in the rate of referrals to hand trauma units after the start of the lockdown in London (75% decrease) and Paris (67% decrease), our results were less dramatic; the maximum monthly decrease in the referral rate during the lockdown was 30%, which was mainly due the reduced rate of fractures [5,6]. It must be noted that the first wave of COVID-19 was much milder in Finland. Our results in adults reflects the previous report over Finnish children, which demonstrated that the rate of operatively treated upper limb traumas remained nearly unchanged during the first wave of the pandemic [7].

The profile and etiology of hand and wrist trauma is likely to vary between countries and demonstrates that country-specific characteristics exist. For example, in Finland the trauma profile varies by the season. The Finnish winter is slippery due to ice and snow, which can be seen in the increase in wrist and hand fractures [11,12]. However, more hand fractures occur among children during warm summer days [13,14]. In recent years, there has been increasing rates of falls from roofs due to snow clearing and a specific type of thumb avulsion injury caused by motorized ice drill accidents in Finland [15,16].

The COVID-19 lockdown was introduced introduced in mid-March. At the time of the lockdown beginning the winter holiday period was over in Southern Finland. The country is relatively large with changing weather and the winter season ends over a month earlier in Southern part than in the Northern. In early April many families start the summer cabin season in the south parts of the Finland and begin do-it-yourself renovations and wood chopping preparings for the next winter [17]. There is no information on the specific incidence of injuries caused by chopping firewood in Finland, but a total of 67 amputation or crush injuries caused by powered log splitters were operated during a two-year period in the TAUH region [18]. In Germany, 80% of saw injuries take place outside of work and half of the injuries are related to cutting firewood [19]. Powered cutting tools are more likely to cause injuries than an axe, which has been reported to relate to only 10% of the hospitalized wood chopping injuries [20].

Although, the overall incidence of hand injuries decreased during the lockdown, the overall rate of emergent and urgent hand operations remained constant during lockdown. Small decrease was seen in February before the lockdown and in March during the lockdown in the incidence of fracture operation, which may be explained by the weather in addition to the lockdown. The possible decrease in the rate of traffic and work-related hand injuries may have been augmented by an increase in the rate of domestic hand injuries and thus explain the unchanged incidence of severe hand injuries. According to a previous Nordic study, 36% of hand injuries occur at home, 36% during leisure activities, and 26% at work [8]. In Finland, however, work-related injuries are rare. According to the Finnish Workers' Compensation

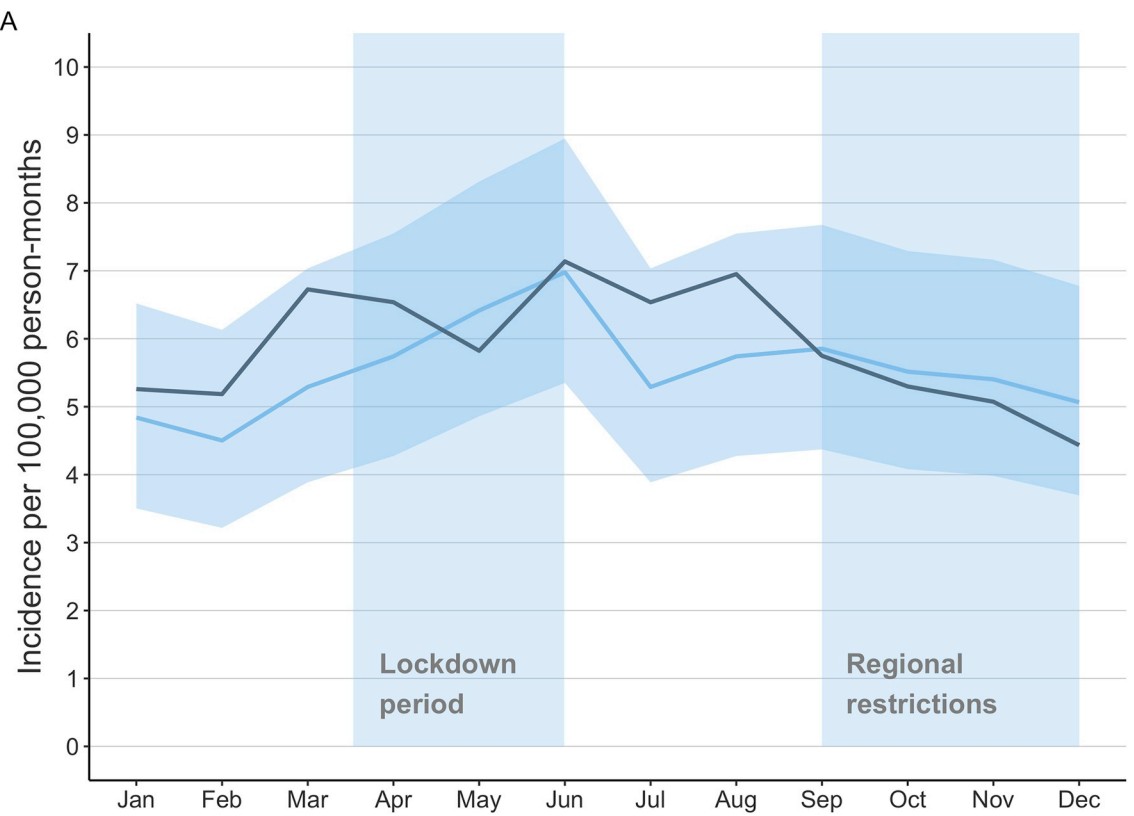

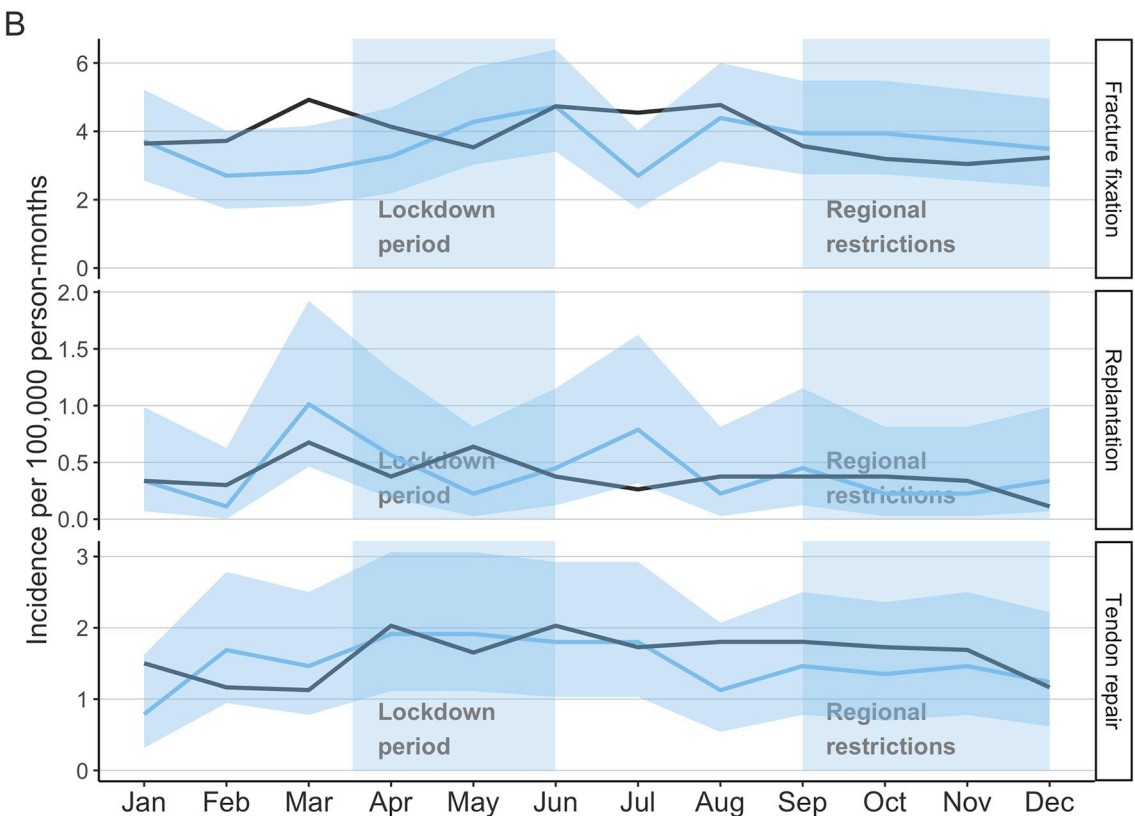

**Fig 2. A** Monthly incidences of emergency and urgent hand operations. Blue line with 95% confidence intervals (light blue) present year 2020 and black line presents the average incidence for reference years 2017–2019. **B** Monthly incidences of emergency and urgent hand operations stratified by the operation type. Blue line with 95% confidence intervals (light blue) present year 2020 and black line presents the average incidence for reference years 2017–2019.

Center, the incidence of work-related traumas in Finland was 29 per million working hours in 2019. Moreover, nearly 80% of injuries leading to absence from work occur outside the workplace, typically at home. In 2020, the incidence of traumatic injuries at work seemed to be lower during the lockdown period in comparison with 2019, which may be explained by the decrease in traffic and other activities with an increased risk of injury [21]. The decrease in traffic also led to a decrease in traffic accidents and traffic deaths from April through to May 2020 compared to the ten previous years [22].

When the lockdown restrictions were lifted in June 2020, the injury patterns returned to normal. People returned to workplaces instead of remote working from home, and restrictions on public institutions and social gathering were eased during the summer due to low pandemic phase. When the second wave began in September 2020 regional restrictions came into effect. The difference, when compared with lockdown, was that regional restrictions were aimed towards the adult population. Operation of restaurants and bars was restricted, remote working was recommended, and hobbies and activities were not banned but some restrictions were enforced. These applied regional restrictions had no observable impact on the incidences of ED visits due hand injuries or emergency hand operations.

The strength of the study was the public and practically free healthcare system in Finland that allows patients to seek medical assistance whenever needed and thus the data reflects most likely true incidences. A minor limitation is the lack of information on private health care facilities, in which a small proportion of minor hand traumas have been treated. However, practically all major traumas and traumas needing acute surgery are treated in public hospitals.

**Table 2. Demographic data of the ED visits due hand injuries and subsequent hand operations, 2020 compared with mean of reference years 2017–2019.** Time stratified by the national lockdown and regional restrictions in effect during the first and second waves of COVID-19 pandemic in Finland.

| | Before Lockdown January to February | | Lockdown March to May | | After lockdown June to August | | Regional restrictions September to December | |
| --- | --- | --- | --- | --- | --- | --- | --- | --- |
| | **2020** | **2017–2019** | **2020** | **2017–2019** | **2020** | **2017–2019** | **2020** | **2017–2019** |
| | **N (%)** | **N (%)** | **N (%)** | **N (%)** | **N (%)** | **N (%)** | **N (%)** | **N (%)** |
| Total ED visits | 1 238 (100) | 1 302 (100) | 1 747 (100) | 2 178 (100) | 2 399 (100) | 2 297 (100) | 2 331 (100) | 1 921 (100) |
| Gender male | 615 (49.7) | 635 (48.7) | 950 (54.4) | 1 099 (50.5) | 1 373 (57.2) | 1 337 (58.2) | 1 233 (52.9) | 1 307 (52.7) |
| Age, mean (SD) | 41.1 (23) | 41.8 (23) | 42.5 (23) | 42.2 (23) | 41.5 (24) | 40.5 (24) | 40.4 (24) | 40.8 (23) |
| ED visit diagnose | | | | | | | | |
| Minor injury | 550 (44.4) | 558 (42.8) | 904 (51.7) | 972 (44.5) | 1 257 (52.4) | 1 193 (52.0) | 1 210 (51.9) | 1 255 (50.6) |
| Fractures and dislocations | 637 (51.5) | 695 (53.3) | 752 (43.0) | 1 100 (50.5) | 1 029 (42.9) | 983 (42.8) | 1 000 (42.9) | 1 115 (45.8) |
| Tendon injury | 26 (2.1) | 27 (2.1) | 41 (2.3) | 53 (2.4) | 46 (1.9) | 64 (2.8) | 49 (2.1) | 62 (2.5) |
| Major injury | 25 (2.0) | 23 (1.8) | 50 (2.9) | 58 (2.7) | 67 (2.8) | 57 (2.4) | 72 (3.1) | 48 (1.9) |
| Total operations | 83 (6.7) | 93 (7.2) | 155 (8.9) | 169 (7.8) | 160 (6.7) | 183 (8.0) | 194 (8.3) | 182 (9.5) |
| Gender male | 62 (74.7) | 58 (62.4) | 110 (71.0) | 117 (69.2) | 129 (80.6) | 137 (74.9) | 135 (69.6) | 129 (70.9) |
| Age, mean (SD) | 37.9 (19) | 42.9 (21) | 45.5 (19) | 44.6 (20) | 41.1 (19) | 40.6 (20) | 40.7 (19) | 41.0 (19) |
| Operation | | | | | | | | |
| Fracture repair | 57 (68.7) | 65 (69.9) | 92 (59.4) | 112 (66.2) | 105 (65.6) | 125 (68.3) | 134 (69.1) | 116 (63.8) |
| Replantation | 4 (4.8) | 4 (4.3) | 16 (10.3) | 15 (8.9) | 13 (8.2) | 9 (4.9) | 11 (5.7) | 10 (5.5) |
| Tendon repair | 22 (26.5) | 24 (25.8) | 47 (30.3) | 42 (24.9) | 42 (26.2) | 49 (26.8) | 49 (25.2) | 56 (30.7) |
| Waiting time from ED to OR <48h | 44 (53.1) | 63 (67.7) | 111 (71.6) | 116 (68.6) | 102 (63.8) | 125 (68.3) | 105 (54.1) | 126 (69.2) |

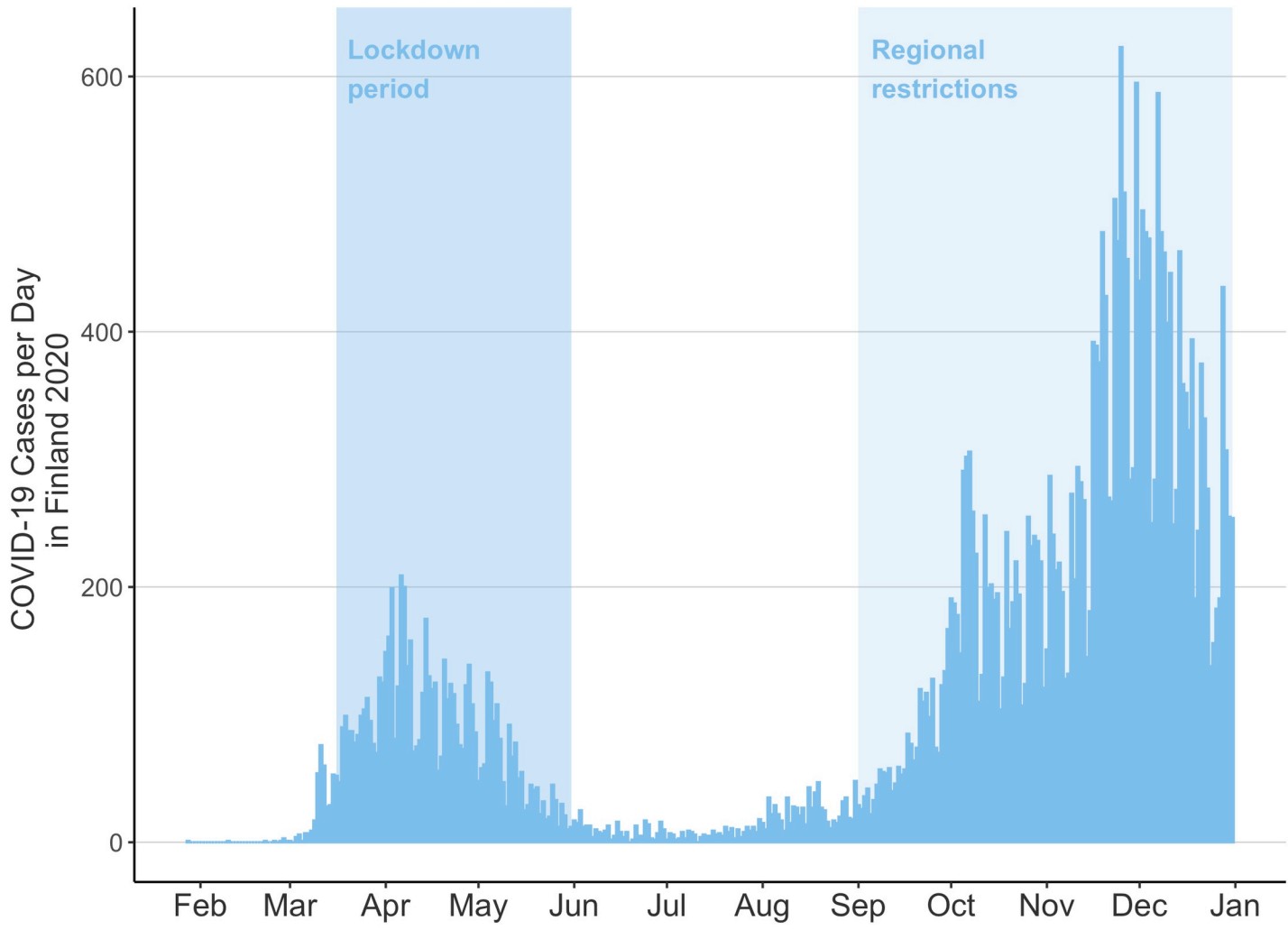

**Fig 3. Daily number of PCR test-positive COVID-19 findings in Finland in 2020.**

Another limitation of our study was that we only had information on the diagnoses and external causes of the visits to the ED or operation, and we were thus unable to classify the injuries by specific etiology and occurring site.

The nationwide COVID-19 lockdown decreased the rate of visits to the ED due to hand injuries in three Finnish hospitals, but the rate of emergency and urgent hand operations remained unchanged during the lockdown. As the restrictions were eased, the visit rates returned to the level of reference years. Enforcing a variety of regional restrictions had no observable impact on the hand injuries during the second wave of COVID-19 in Finland. Most likely this was due to less strict restrictions and therefore, for example, many employers were back to workplaces, sports facilities were mainly open, and domestic traveling was more accepted, whereas during the lockdown all of these were recommended against. Compared to other countries Finland had one of the lowest daily numbers of positive cases in 2020 and this is most likely a contributing factor to these results, as the daily numbers most likely reflect to behavior and risk taking, and Finland did not need a curfew, which most likely would have reduced the trauma rates.

The identification of regional changes in injury patterns and the subsequent demand for surgical care are important factors in ensuring the effective treatment of hand injuries in the ED and operation room. The findings of this study will therefore provide better tools for the planning of the optimal and sustainable use of resources during a future national state of emergency.

## Acknowledgments

We would like to thank Mr Peter Heath for language corrections.

## Author Contributions

**Conceptualization:** Jarkko Jokihaara, Aleksi Reito, Ville M. Mattila, Antti P. Launonen.

**Data curation:** Ilari Kuitunen, Ville Ponkilainen, Aleksi Reito.

**Formal analysis:** Ilari Kuitunen, Ville Ponkilainen, Antti P. Launonen.

**Funding acquisition:** Juha Paloneva, Ville M. Mattila.

**Investigation:** Ilari Kuitunen, Jarkko Jokihaara, Aleksi Reito, Juha Paloneva, Ville M. Mattila, Antti P. Launonen.

**Methodology:** Ilari Kuitunen, Ville Ponkilainen, Aleksi Reito, Ville M. Mattila.

**Project administration:** Jarkko Jokihaara, Juha Paloneva, Ville M. Mattila, Antti P. Launonen.

**Resources:** Juha Paloneva, Ville M. Mattila, Antti P. Launonen.

**Software:** Ville Ponkilainen, Aleksi Reito, Ville M. Mattila.

**Supervision:** Jarkko Jokihaara, Juha Paloneva, Ville M. Mattila, Antti P. Launonen.

**Validation:** Ville Ponkilainen, Aleksi Reito, Juha Paloneva, Ville M. Mattila, Antti P. Launonen.

**Visualization:** Ville Ponkilainen.

**Writing – original draft:** Ilari Kuitunen.

**Writing – review & editing:** Ilari Kuitunen, Jarkko Jokihaara, Ville Ponkilainen, Aleksi Reito, Juha Paloneva, Ville M. Mattila, Antti P. Launonen.

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
