## [Decision Letter · Decision Letter 0]

8 Oct 2021

PONE-D-21-10154Emergency department visits due to hand trauma and subsequent emergency hand surgery in three Finnish hospitals during the first and second waves of COVID-19 pandemicPLOS ONE

Dear Dr. Kuitunen,

Thank you for submitting your manuscript to PLOS ONE. After careful consideration, we feel that it has merit but does not fully meet PLOS ONE’s publication criteria as it currently stands. Therefore, we invite you to submit a revised version of the manuscript that addresses the points raised during the review process.

We look forward to receiving your revised manuscript.

Kind regards,

Gabriel de Araújo, M.D., MSc

Academic Editor

PLOS ONE

Journal Requirements:

Reviewers' comments:

Reviewer's Responses to Questions

**Comments to the Author**

1. Is the manuscript technically sound, and do the data support the conclusions?

Reviewer #1: Yes

Reviewer #2: Yes

2. Has the statistical analysis been performed appropriately and rigorously? 

Reviewer #1: Yes

Reviewer #2: Yes

3. Have the authors made all data underlying the findings in their manuscript fully available?

Reviewer #1: Yes

Reviewer #2: Yes

4. Is the manuscript presented in an intelligible fashion and written in standard English?

Reviewer #1: Yes

Reviewer #2: Yes

5. Review Comments to the Author

Reviewer #1: Although the number of emergency outpatients was reduced due to the lockdown, the number of surgeries did not seem to have changed significantly compared to before the pandemic. Since all patients can receive medical care in Finland's medical system, the strength of this study is that the numbers given in this study can be regarded as actual numbers.

On the other hand, compared to reports from other countries such as the United Kingdom and France, there seems to be little change in the number of emergency outpatients and the number of surgeries. This is thought to be affected by the number of people infected with COVID-19. In other words, since the number of COVID-19-infected persons is smaller in Finland than in France and the United Kingdom, it is considered that this pandemic had less impact on society such as emergency outpatient visits and surgery.

In this study, lockdown and regional restrictions are clarified in the figure, but it is easy for the reader to understand that they also include data on changes in the number of infected people in Finland.

Reviewer #2: This is an interesting and important research of the effect of COVID19 to hand trauma in Finland. Despite being a regional study, the enrollment is satisfactory in making generalization. As main findings, the researchers report that hand traumas decreased in the national lockdown, but thereafter increased to the reference level despite the later regional restrictions. Another important finding was that national lockdown didn't decrease the emergency operations.

The study is impotant and it adds the literature. The study is in the scope of the Journal. Language is good, the structure, figures and tables are sound. However, there are some minor point that I hope the authors would still consider for the revision:

- you conclude that all hand traumas have been treated in public hospitals. Perhaps some of them have been treated in private hospitals? This should be discussed in the manuscript, albeit it will not change the main findings and the value of the study.

- it is unclear, if the entire population, including children and adolescents, are included. There is variation in the study country regarding the in-hopsital treatment of childhood hand trauma; are they admitted to the dep. of pediatric surgery and orthopaedics or to the dep. of hand surgery. Please, define that in the revision.

- There has been one previous study of childhood trauma (including hand) in pediatric patients in the study country (A. Raitio et al, SJS 2020), which should be cited, to my opinion.

- Regarding the results, I think the incidence of replantantion emergency operations didn't change, while looking to the great confidence intervals of the IRRs. Therefore, the results and the conclusion concerning the change of IRR of replantantions should be re-written more carefully.

- It sounds little odd that summer cabin season starts in March in Finland. At the time of the first lockdown, the winter holidays were just starting. Perhaps some clarification about the large country and the potential variation in the study country, between the northern and southern parts of the country, could be added.

- As a part of final conclusions, I would like to see little more deductive speculation, why regional restrictions didn't decrease hand trauma.

In conclusion, this is very interesting study and I suggest publication after abovementioned minor repairs.

6. PLOS authors have the option to publish the peer review history of their article (what does this mean?). If published, this will include your full peer review and any attached files.

Reviewer #1: No

Reviewer #2: **Yes: **Prof. Juha-Jaakko Sinikumpu

---

## [Author Response · Author response to Decision Letter 0]

16 Oct 2021

Reviewer #1: Although the number of emergency outpatients was reduced due to the lockdown, the number of surgeries did not seem to have changed significantly compared to before the pandemic. Since all patients can receive medical care in Finland's medical system, the strength of this study is that the numbers given in this study can be regarded as actual numbers.

On the other hand, compared to reports from other countries such as the United Kingdom and France, there seems to be little change in the number of emergency outpatients and the number of surgeries. This is thought to be affected by the number of people infected with COVID-19. In other words, since the number of COVID-19-infected persons is smaller in Finland than in France and the United Kingdom, it is considered that this pandemic had less impact on society such as emergency outpatient visits and surgery.

In this study, lockdown and regional restrictions are clarified in the figure, but it is easy for the reader to understand that they also include data on changes in the number of infected people in Finland.

Author answer: We would like to thank reviewer 1 for these comments. 

Reviewer #2: This is an interesting and important research of the effect of COVID19 to hand trauma in Finland. Despite being a regional study, the enrollment is satisfactory in making generalization. As main findings, the researchers report that hand traumas decreased in the national lockdown, but thereafter increased to the reference level despite the later regional restrictions. Another important finding was that national lockdown didn't decrease the emergency operations.

The study is impotant and it adds the literature. The study is in the scope of the Journal. Language is good, the structure, figures and tables are sound. However, there are some minor point that I hope the authors would still consider for the revision: 

- you conclude that all hand traumas have been treated in public hospitals. Perhaps some of them have been treated in private hospitals? This should be discussed in the manuscript, albeit it will not change the main findings and the value of the study.

Author answer: Thank you for this important point. We have now revised the sentence and included this as a limitation for our study (lines 186-189). 

- it is unclear, if the entire population, including children and adolescents, are included. There is variation in the study country regarding the in-hopsital treatment of childhood hand trauma; are they admitted to the dep. of pediatric surgery and orthopaedics or to the dep. of hand surgery. Please, define that in the revision.

Author answer: Again, an excellent point which needs to be clarifed. We included only patients aged 16 or more. This has been added to methods (lines 77-78). 

- There has been one previous study of childhood trauma (including hand) in pediatric patients in the study country (A. Raitio et al, SJS 2020), which should be cited, to my opinion.

Author answer: Thank you for this suggestion. We have now cited this study (reference 7) and added this in introduction and discussion (lines 59-61 and 142-144).

- Regarding the results, I think the incidence of replantantion emergency operations didn't change, while looking to the great confidence intervals of the IRRs. Therefore, the results and the conclusion concerning the change of IRR of replantantions should be re-written more carefully.

Author answer: We have now revised the interpretation of replantations, please see abstract (line 35), results (line 125) and discussion (lines 134 and 138).

- It sounds little odd that summer cabin season starts in March in Finland. At the time of the first lockdown, the winter holidays were just starting. Perhaps some clarification about the large country and the potential variation in the study country, between the northern and southern parts of the country, could be added.

Author answer: This is a good point and we have now added more information and context to discussion (lines 151-155). 

- As a part of final conclusions, I would like to see little more deductive speculation, why regional restrictions didn't decrease hand trauma.

Author answer: We have updated the conclusion section as suggested (lines 196-198). 

In conclusion, this is very interesting study and I suggest publication after abovementioned minor repairs.

Author answer: We would like to thank reviewer 2 for the kind words and excellent suggestions!

---

## [Decision Letter · Decision Letter 1]

7 Dec 2021

PONE-D-21-10154R1Emergency department visits due to hand trauma and subsequent emergency hand surgery in three Finnish hospitals during the first and second waves of COVID-19 pandemicPLOS ONE

Dear Dr. Kuitunen,

Thank you for submitting your manuscript to PLOS ONE. After careful consideration, we feel that it has merit but does not fully meet PLOS ONE’s publication criteria as it currently stands. Therefore, we invite you to submit a revised version of the manuscript that addresses the points raised during the review process.

We look forward to receiving your revised manuscript.

Kind regards,

Gabriel de Araújo, M.D., MSc

Academic Editor

PLOS ONE

Journal Requirements:

Additional Editor Comments (if provided):

Dear Author,

Your manuscript is good for publication, however the reviewer signed that the requested modification was not added.

I am waiting for your revision for a final decision.

Best regards,

Gabriel de Araújo

Reviewers' comments:

Reviewer's Responses to Questions

**Comments to the Author**

1. If the authors have adequately addressed your comments raised in a previous round of review and you feel that this manuscript is now acceptable for publication, you may indicate that here to bypass the “Comments to the Author” section, enter your conflict of interest statement in the “Confidential to Editor” section, and submit your "Accept" recommendation.

Reviewer #1: (No Response)

Reviewer #2: All comments have been addressed

2. Is the manuscript technically sound, and do the data support the conclusions?

Reviewer #1: Partly

Reviewer #2: Yes

3. Has the statistical analysis been performed appropriately and rigorously? 

Reviewer #1: Yes

Reviewer #2: Yes

4. Have the authors made all data underlying the findings in their manuscript fully available?

Reviewer #1: Yes

Reviewer #2: Yes

5. Is the manuscript presented in an intelligible fashion and written in standard English?

Reviewer #1: (No Response)

Reviewer #2: Yes

6. Review Comments to the Author

Reviewer #1: It is reported that the impact of COVID-19 on the hand trauma of the emergency outpatient clinic in Finland was small. Since the number of people infected with COVID-19 in Finland is smaller than in other countries, the impact of pandemic on daily life seems to be suppressed.

I commented last time that the number of COVID-19 infected people in Finland during study period should be included for reader to understand easily, but no data has been added.

Reviewer #2: I'm happy with the revision the authors have made. I recommend editor to accept the manuscript in this form.

7. PLOS authors have the option to publish the peer review history of their article (what does this mean?). If published, this will include your full peer review and any attached files.

Reviewer #1: No

Reviewer #2: No

---

## [Author Response · Author response to Decision Letter 1]

9 Dec 2021

Reviewer #1: It is reported that the impact of COVID-19 on the hand trauma of the emergency outpatient clinic in Finland was small. Since the number of people infected with COVID-19 in Finland is smaller than in other countries, the impact of pandemic on daily life seems to be suppressed.

I commented last time that the number of COVID-19 infected people in Finland during study period should be included for reader to understand easily, but no data has been added.

Author answer: Thank you for this suggestion. We have now included Figure 3, which includes daily number of positive COVID-19 cases in Finland in 2020. We have updated the methods, results, discussion and figure legends section now.

---

## [Editor Report · Decision Letter 2]

20 Jan 2022

Emergency department visits due to hand trauma and subsequent emergency hand surgery in three Finnish hospitals during the first and second waves of COVID-19 pandemic

PONE-D-21-10154R2

Dear Dr. Kuitunen,

We’re pleased to inform you that your manuscript has been judged scientifically suitable for publication and will be formally accepted for publication once it meets all outstanding technical requirements.

Kind regards,

Gabriel de Araújo, M.D., MSc

Academic Editor

PLOS ONE

---

## [Editor Report · Acceptance letter]

24 Jan 2022

PONE-D-21-10154R2 

Emergency department visits due to hand trauma and subsequent emergency hand surgery in three Finnish hospitals during the first and second waves of COVID-19 pandemic 

Dear Dr. Kuitunen:

I'm pleased to inform you that your manuscript has been deemed suitable for publication in PLOS ONE. Congratulations! Your manuscript is now with our production department. 

Kind regards, 

on behalf of

Professor Gabriel de Araújo 

Academic Editor

PLOS ONE